# Single-Center Experience of Parathyroidectomy Using Intraoperative Parathyroid Hormone Monitoring

**DOI:** 10.3390/medicina58101464

**Published:** 2022-10-16

**Authors:** Seong Hoon Kim, Si Yeon Lee, Eun Ah Min, Young Mi Hwang, Yun Suk Choi, Jin Wook Yi

**Affiliations:** Department of Surgery, Inha University Hospital, College of Medicine, Incheon 22332, Korea

**Keywords:** parathyroid hormone, hyperparathyroidism, intraoperative parathyroid hormone

## Abstract

*Background and Objectives*: Hyperparathyroidism (HPT) is a rare endocrine disease associated with the elevated metabolism of calcium, vitamin D, and phosphate by the hyperfunctioning of the parathyroid glands. Here, we report our experience of parathyroidectomy using intraoperative parathyroid hormone (IOPTH) monitoring in a single tertiary hospital. *Materials and Methods*: From October 2018 to January 2022, a total of 47 patients underwent parathyroidectomy for HPT. We classified the patients into two groups—primary HPT (PHPT, n = 37) and renal HPT (RHPT, n = 10)—and then reviewed the patients’ data, including their general characteristics, laboratory results, and perioperative complications. *Results*: Thirty-five of the thirty-seven patients in the PHPT group underwent focused parathyroidectomy, while all ten patients in the RHPT group underwent subtotal parathyroidectomy. IOPTH monitoring based on the Milan criteria was used in all cases. Preoperative and 2-week, 6-month, and 12-month postoperative parathyroid hormone (PTH) levels were within the normal range in the PHPT group, whereas they were higher than normal in the RHPT group. Transient hypocalcemia occurred only in the RHPT group, with calcium levels returning to normal levels 12 months after surgery. *Conclusions*: Parathyroidectomy with IOPTH monitoring in our hospital showed favorable clinical outcomes. However, owing to the small number of patients due to the low frequency of parathyroid disease, long-term, prospective studies are needed in the future.

## 1. Introduction

Hyperparathyroidism (HPT) is an endocrine disease related to the elevated metabolism of calcium (Ca), vitamin D, and phosphate (P) due to the excessive secretion of parathyroid hormone (PTH) [1,2,3]. HPT is classified into primary (PHPT), secondary (SHPT), and tertiary (THPT) disease depending on the disease mechanism. HPT raises serum Ca and P levels and lowers vitamin D levels, resulting in various diseases, including bone, kidney, gastrointestinal, neuropsychiatric, soft tissue, and cardiovascular disorders. A lack of proper treatment of the disease associated with HPT, especially cardiovascular complications, may lead to fatal consequences for the patient [1,2,3]. The treatment required for PHPT is surgical resection of the pathologic parathyroid gland; some medically intractable SHPT and THPT patients also require surgical treatment [4,5]. 

PHPT is a disease caused by the excessive production of PTH from one or more parathyroid glands. The incidence of PHPT is reported to be approximately 20–50 cases per 100,000 people annually, and it is the third-most common endocrine disorder in the United States [6,7]. Owing to the development of various diagnostic modalities, the number of cases of PHPT has been increasing [8]. Approximately 85% of PHPT cases result from a single parathyroid adenoma, while the remaining 15% of the cases are caused by multiple abnormal glands [9]. In the case of PHPT, surgical excision rather than pharmacological therapy is recommended as the gold standard treatment. Previously, surgery was performed to examine all parathyroid glands using a large incision, but focused parathyroidectomy using minimal skin incision has been introduced with the development of preoperative localizing imaging techniques and intraoperative PTH (IOPTH) monitoring to confirm the complete excision of the pathologic parathyroid glands [10,11,12].

SHPT is a condition characterized by elevated PTH secretion caused by abnormal vitamin D and calcium metabolism. The major cause of SHPT is chronic renal failure, but other causes include gastrointestinal absorption dysfunction, vitamin D deficiency, liver diseases, and pseudohypoparathyroidism [4,5]. THPT is a condition of persistent HPT after kidney transplantation [1,12]. SHPT and THPT are both caused by renal problems and can be classified as RHPT. In most cases of RHPT, pharmacological therapy using cinacalcet is recommended as the initial treatment. When this pharmacological therapy fails to produce a proper response, parathyroidectomy—total parathyroidectomy with autotransplantation or subtotal parathyroidectomy—should be recommended [13,14].

Because the incidence of HPT is very low, parathyroid surgery is performed in very few institutions in Korea [12,13,15,16,17,18,19,20]. There are even fewer centers that use the IOPTH assay for monitoring for parathyroid surgery in Korea [21,22,23,24]. A single endocrine surgeon at our hospital has been performing parathyroid surgery using IOPTH since 2018, and we report our initial results of parathyroid surgery in this paper. 

## 2. Materials and Methods

### 2.1. Patients

From September 2018 to January 2022, a total of 47 patients received parathyroid surgery due to HPT. All patients who underwent parathyroidectomy at Inha University were included, and there were no special exclusion criteria. The clinical diagnoses were as follows: PHPT in 37 patients, SHPT in 9 patients, and THPT in 1 patient. In the analysis, the SHPT and THPT patients were included in the RHPT group as they shared the same origin of HPT (renal dysfunction). We retrospectively reviewed the electronic medical records for the patients’ clinical information, laboratory test results (for Ca, ionized Ca, P, and PTH), surgical findings, hospitalization records, and pathologic reports.

### 2.2. Surgical Indication

In PHPT, surgery was performed on patients who had symptoms (nephrolithiasis, fractures, symptomatic hypercalcemia). For asymptomatic PTHP patients, surgery was performed if serum calcium was more than 1.0 mg/dL above the normal, creatinine clearance <60 cc/min, nephrocalcinosis or nephrolithiasis identified on imaging, 24 h urine calcium >400 mg/day, osteoporosis by bone density at any site (T score ≤−2.5), clinical fragility fracture, vertebral compression fracture on spine imaging, or age <50 years. If medical therapy was refractory, surgery was performed on SHPT patients with symptoms and markedly elevated PTH levels. That includes a persistent high serum level of intact PTH level >500 pg/m, hyperphosphatemia (serum P > 6.0 mg/dL) and/or hypercalcemia (serum Ca > 10.0 mg/dL), deformity, fracture, progressive reduction in bone mineral content, ectopic calcification, or neuro-muscular psychiatric symptoms, etc. In THPT, surgery was performed on patients with persistent hypercalcemia more than 6 months after kidney transplant, low bone mineral density, renal stone or nephrocalcinosis, deterioration of kidney graft due to THPT, symptomatic HPT, or enlarged parathyroid gland detected by the US [2,3].

### 2.3. Surgery Methods and IOPTH Monitoring

All surgeries were performed by a single endocrine surgeon (JW Yi). A 2–3 cm skin incision was used to perform focused parathyroidectomy for PHPT. Before the surgery, the location of the pathologic parathyroid gland was determined by parathyroid SPECT-CT and ultrasound-guided skin marking. Subtotal parathyroidectomy (the removal of three and a half parathyroid glands) was applied for RHPT patients using a transverse skin incision of ~7–9 cm. Intraoperative neuromonitoring (NIM 3.0, Medtronic, Minneapolis, MN, USA) was used in all parathyroid surgeries.

The IOPTH assay was performed at four surgical time points: pre-incision, pre-excision, and 5 min and 10 min after the removal of the pathologic parathyroid gland. The Miami criterion (a >50% decrease in PTH 10 min after parathyroidectomy compared to the highest level of PTH before the excision) was indicated for the successful removal of the hyperfunctioning parathyroid gland in both focused and subtotal parathyroidectomy in our institution [25]. In focused parathyroidectomy, bilateral exploration was indicated when the PTH level was not successfully decreased. For the subtotal parathyroidectomy, all parathyroid glands were identified through bilateral exploration, saving half of the most normal-looking gland and removing the remaining three glands. Pre-excision PTH was sampled after one and a half glands were removed, while post-excision PTH was sampled 5 and 10 min after the complete removal of the three and a half glands. Laboratory tests for PTH, Ca, ionized Ca, and P were performed on the admission day, the day after surgery, and 6 and 12 months after surgery. 

### 2.4. Statistics and Ethical Consideration

We used R programming language version 4.2.0 for the statistical analysis [15]. Chi-square or Fisher’s exact test was applied to the cross-table analysis according to the sample size. The unpaired t-test was used for the mean comparison. Statistical significance was defined as a *p*-value < 0.05. Ethical approval for this study was obtained from the institutional review board of our hospital (IRB number: 2022-05-028).

## 3. Results

Table 1 presents the clinical and pathologic characteristics of patients. Among all 47 patients, 37 were female and 10 were male; the mean age was 56.50 ± 11.54 years. The clinical diagnoses were PHPT in 37 patients, SHPT in 9 patients, and THPT in 1 patient. The pathologic diagnoses were parathyroid adenoma and parathyroid hyperplasia in 31 and 16 patients, respectively. Among the 37 PHPT patients, one-gland parathyroidectomy was performed in 35 patients, two-gland parathyroidectomy was performed in 1 patient, and bilateral exploration with the removal of three glands was performed in 1 patient. Subtotal parathyroidectomy was performed in all 10 patients with RHPT. Although transient vocal cord palsy occurred in two patients, no permanent vocal cord palsy was observed. Hypertrophic scars were observed in two patients.

Table 2 presents the comparative results between PHPT and RHPT. Age did not differ between the two groups, but the proportion of female patients was significantly higher in the PHPT group than in the RHPT group (32/37 (86.5%) versus 5/10 (50.0%); *p* = 0.024). The major clinical manifestations of PHPT were ureteral or renal stones, osteoporosis, and a history of bone fracture. The operation time was longer (68.2 ± 42.8 min versus 83.0 ± 21.4 min, *p* = 0.139), the estimated blood loss was higher (10.8 ± 20.1 mL versus 130.0 ± 248.6 mL, *p* = 0.164), and the hospital stay days were longer (1.8 ± 1.1 days versus 6.0 ± 6.2 days, *p* = 0.059) in the RHPT group. In the PHPT group, the most common positions of the hyperfunctioning glands were the lower right (14 glands) and lower left (12 glands). The upper left was the most common location of the saved glands in the RHPT group. 

Table 3 and Figure 1 and Figure 2 show the laboratory findings of the PHPT and RHPT groups. The preoperative PTH level was significantly higher in the RHPT group (161.6 ± 95.4 pg/mL versus 1242.1 ± 1075.3 pg/mL; *p* = 0.011). The PTH level measured at 6 months and 12 months after surgery did not differ significantly between the two groups. According to the IOPTH assay results, the PTH levels at all time points were higher in the RHPT group than in the PHPT group. A >50% decrease in the PTH level was observed 10 min after the excision in all patients in the PHPT and RHPT groups. The preoperative Ca level was significantly higher in the PHPT group (10.9 ± 0.9 mg/dL versus 9.6 ± 1.7 mg/dL; *p* = 0.042). After surgery, the Ca level was normalized in the PHPT group, whereas hypocalcemia was observed in the RHPT group at 6 months after surgery (9.5 ± 1.3 mg/dL versus 6.8 ± 1.2 mg/dL; *p* = 0.003). No difference was observed in the Ca levels 12 months after surgery (8.6 ± 2.8 mg/dL versus 8.6 ± 2.07 mg/dL; *p* = 0.942). There was no significant difference in the ionized Ca level between the two groups before or after surgery. The P level was significantly higher in the RHPT group before surgery (2.6 ± 0.5 mg/dL versus 3.9 ± 1.1 mg/dL; *p* = 0.023) but decreased and increased again 12 months after surgery, without any significant difference between the PHPT and RHPT groups (3.3 ± 0.6 mg/dL versus 4.1 ± 1.3 mg/dL; *p* = 0.377).

## 4. Discussion

In 1925, in what is known as the first attempted surgical treatment of HPT, Felix Man reported that clinical symptoms improved after removing the enlarged parathyroid gland in a patient with severe bone lesions [26]. With the development of automatic analyzers that can measure serum Ca levels, the detection rate of hypercalcemia has increased since the 1970s. Previously, PHPT was mainly diagnosed by characteristic symptoms, but recently, asymptomatic HPT has been mainly diagnosed through health examinations using biochemical analyses and neck imaging techniques, including ultrasound and computed tomography [27]. According to a single-center experience in Korea, there have been more HPT patients during the last 6 years than during the previous 20 years [28]. This means that the diagnosis rate of HPT patients is increasing in Korea with the development of health screening. However, according to national health insurance data, the incidence of PHPT in Korea is still very low, with an annual incidence ranging from 0.007% to 0.0014% [29]. The most important laboratory tests for HPT are serum Ca and PTH levels. Traditionally, 24 h urine Ca amount, serum chloride concentration, serum chloride-to-P ratio, serum alkaline phosphatase, and tubular P reabsorption rate have been used for differential diagnosis. However, anatomical and functional imaging techniques are more useful in recent clinical diagnosis and surgical planning [30]. Neck ultrasound and 99mTC-sestamibi scanning are very helpful to localize the hyperfunctioning glands, as the sensitivity of ultrasound is 59–89% and that of 99mTC-sestamibi scanning is 54–88% [31,32,33,34,35]. Moreover, the success rate of parathyroid localization increases by 10–20% when both methods are implemented together [36]. The IOPTH assay, which was first proposed in 1988 by Nussbaum et al., is very helpful to confirm the complete removal of the hyperfunctioning parathyroid gland during surgery [37]. It helps the surgeon to evaluate whether bilateral neck exploration is required after focused parathyroidectomy [34,37]. By using localizing imaging techniques and IOPTH, minimally invasive focused parathyroidectomy can be safely achieved for PHPT [36,38,39]. 

According to the National Institutes of Health guidelines, surgical treatment is recommended for all symptomatic PHPT patients and for asymptomatic patients under 50 years of age who meet the following criteria: 24 h urinary Ca excretion ≤ 300 mg, serum Ca more than 1 mg/dL than normal level, creatinine clearance < 30%, or osteoporosis diagnosed by the Dual Energy X-ray Absorptiometry (DEXA) test [40]. The bilateral exploration of the parathyroid glands performed in the past has been recently replaced with focused parathyroidectomy as the standard treatment using localization and IOPTH [10,11,41]. Focused parathyroidectomy has the advantages of a smaller incision, a faster recovery period, and a shorter operation time. In this study, out of the 37 patients in the PHPT group, 35 received focused one-gland parathyroidectomy, 1 received two-gland parathyroidectomy, and 1 received bilateral exploration with the excision of three glands. For the last 2 patients, an extended skin excision was performed because the initial postoperative PTH level did not decrease after the one-gland excision. The PTH level decreased after the additional excision of the parathyroid glands, and the surgery was completed. Thus, our experience suggests that IOPTH is useful in preventing reoperation in some PHPT patients (2/37 (5.4%)).

On the other hand, SHPT is a disease in which the observed parathyroid hyperplasia is due to a damaged kidney. Approximately 5–25% of chronic renal failure patients require surgical treatment of their SHPT [42,43]. According to the Japanese Society of Dialysis Therapy (JSDT) guidelines, the target laboratory ranges for P, Ca, and PTH in SHPT patients are 3.5–6.0 mg/dL, 8.4–10.0 mg/dL, and 60–180 pg/mL, respectively [44]. Pharmacological therapy—vitamin D2, D3 analogs, and calcium preparations—is generally recommended as the initial therapy [41,45,46,47]. However, the Kidney Disease Outcomes Quality Initiative (KDOQI) guideline recommends that parathyroidectomy should be considered when the PTH level reaches 800 pg/mL, or when the patient suffers from medically intractable hypercalcemia and hyperphosphatemia [48].

Pathologic findings in RHPT are parathyroid hyperplasia in all the parathyroid glands. Thus, the proper removal of the parathyroid glands is very important in RHPT surgery. Therefore, IOPTH is also helpful to confirm the successful removal of the pathologic parathyroid glands in RHPT surgery. Subtotal parathyroidectomy and total parathyroidectomy with autotransplantation are widely used in RHPT surgery. Our center performs subtotal parathyroidectomy due to its several benefits such as shorter operation time, shorter hospital stay, and the reduced need for vitamin D medication [13,14]. As some ectopic parathyroid glands may exist, preoperative images and IOPTH levels are helpful to identify them. In this study, there were no suspected ectopic parathyroid glands in any of the RHPT patients. At the 1-year follow-up, the PTH and Ca levels were stabilized in our patients after the surgery, as shown in Table 3. 

Previous studies have reported that the recurrence rate after parathyroidectomy was 0.4% in PHPT and 26% in SHPT [45,49]. In our study, RHPT patients had significantly higher PTH levels before surgery than PHPT patients; the PTH levels were also significantly higher in the IOPTH assay. The PTH levels in the RHPT group were also high at 6 months and 12 months after surgery. However, the postoperative PTH levels met the JSDT guideline in both groups [44]. The preoperative Ca levels were significantly higher in the PHPT group, but postoperative transient hypocalcemia was observed only in the RHPT group 6 months after surgery (9.5 ± 1.3 mg/dL versus 6.8 ± 1.2 mg/dL; *p* < 0.003). The Ca levels were recovered to normal levels at 12 months after surgery. This result shows that proper postoperative Ca supplements are required for RHPT patients to prevent symptoms of hypocalcemia due to the excessive bone reabsorption of Ca after parathyroidectomy [13,15].

This study reported on parathyroid surgery for the past 2 years at a single center in South Korea. Considering the low incidence of HPT, our sample of 47 patients was meaningful to evaluate the outcome of parathyroid surgery using IOPTH. Although there are studies suggesting that preoperative localization imaging is a powerful tool for the successful resection of pathologic parathyroid glands in experienced centers, the addition of the IOPTH assay should be considered to prevent surgical failure for novice endocrine surgeons or inexperienced hospitals [50].

Our study was a retrospective descriptive analysis with a small number of patients from a single hospital. Considering the low incidence of HPT, we postulate that this study has clinical significance to the endocrine surgery field, especially in South Korea. Nationwide studies should be conducted in future research to evaluate the outcome of parathyroid surgery in South Korea.

## 5. Conclusions

HPT is a disease with a low incidence rate, but the frequency of detection is increasing with the development of medical examination techniques. Focused parathyroidectomy with PHPT is considered acceptable. In RHPT, it is necessary to completely remove the hyperfunctioning gland. The IOPTH assay can help surgeons to reduce incomplete excision and recurrence.

## Figures and Tables

**Figure 1 medicina-58-01464-f001:**
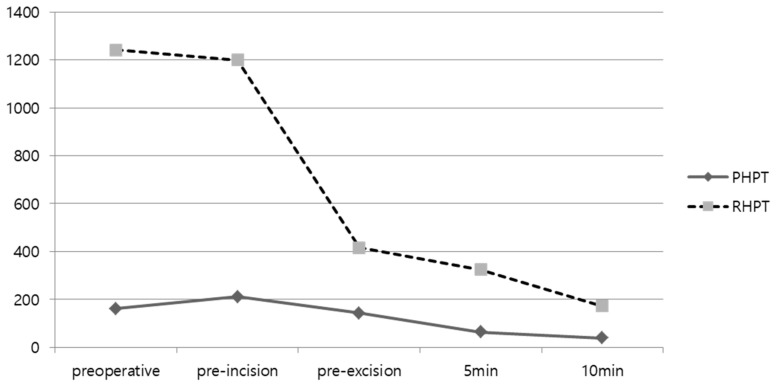
IOPTH assay according to primary hyperparathyroidism (PHPT) and renal (secondary and tertiary) hyperparathyroidism (RHPT).

**Figure 2 medicina-58-01464-f002:**
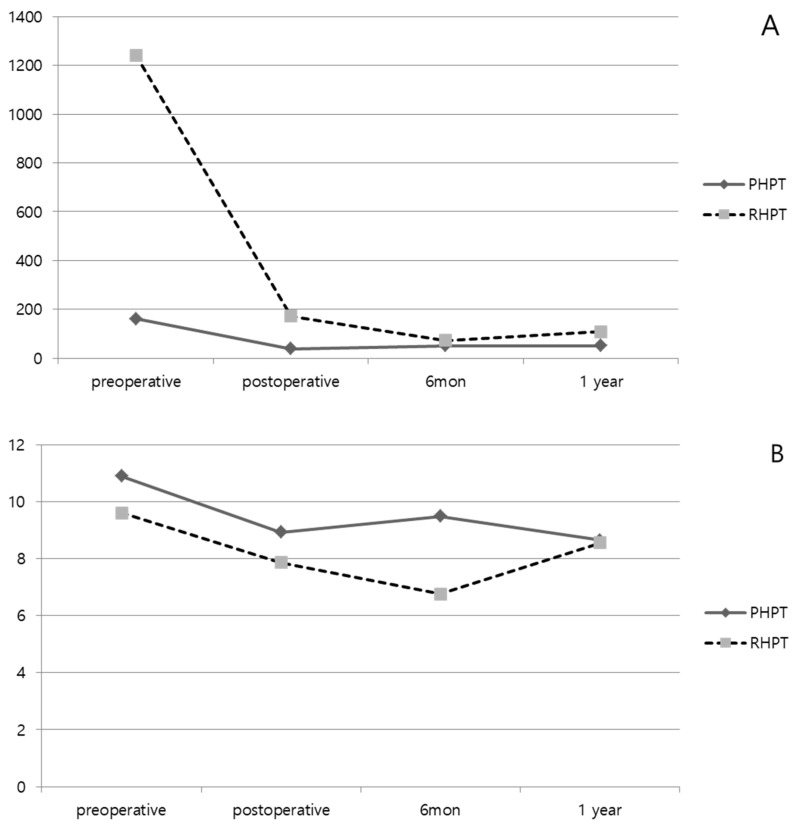
Laboratory results change before and after parathyroidectomy according to primary hyperparathyroidism (PHPT) and renal (secondary and tertiary) hyperparathyroidism (RHPT). ((**A**) PTH. (**B**) Calcium. (**C**) Ionized Calcium. (**D**) Phosphate).

**Table 1 medicina-58-01464-t001:** General characteristics of all hyperparathyroidism patients (n = 47).

Variables	Number of Patients
Age (years, mean ± sd)	56.5 ± 11.5
Sex	
Male	10
Female	37
Clinical diagnosis	
Primary hyperparathyroidism	37
Secondary hyperparathyroidism	9
Tertiary hyperparathyroidism	1
Parathyroidectomy extent	
One-gland parathyroidectomy	35
Two-gland parathyroidectomy	1
Bilateral exploration	1
Subtotal parathyroidectomy	10
Pathologic diagnosis	
Parathyroid adenoma	31
Parathyroid hyperplasia	16
Postoperative complications	
Transient vocal cord palsy	2
Permanent vocal cord palsy	0
Hypertrophic scar or keloid	2

**Table 2 medicina-58-01464-t002:** Comparison between primary and renal (secondary and tertiary) hyperparathyroidism.

Variables	PHPT (n = 37)	RHPT (n = 10)	*p*-Value
Age (years, mean ± sd)	57.1 ± 12.1	54.2 ± 9.3	0.425
Sex			
Male	5 (13.5%)	5 (50.0%)	0.024
Female	32 (86.5%)	5 (50.0%)	
BMI (kg/m^2^, mean ± sd)	24.0 ± 4.0	24.0 ± 2.5	0.983
Comorbidity (number of patients)			
Diabetes	4	1	
Hypertension	9	7	
Chronic renal failure	0	10	
Coronary artery disease	2	0	
Arrhythmia	1	0	
Cerebrovascular disease	1	0	
Hepatitis B	1	0	
Tuberculosis	1	0	
Osteoporosis	4	0	
Fracture history	1	0	
Ureter or renal stone	7	0	
Operation time (min, mean ± sd)	68.2 ± 42.8	83.0 ± 21.4	0.139
Estimated blood loss (mL, mean ± sd)	10.8 ± 20.1	130.0 ± 248.6	0.164
Largest gland size (cm, mean ± sd)	1.8 ± 1.0	1.4 ± 1.1	0.407
Hospital stay days after surgery (days, mean ± sd)	1.8 ± 1.1	6.0 ± 6.2	0.059
Pathologic gland location (Focused parathyroidectomy), number (%)			
Lower right	14 (36.8%)		
Upper right	5 (13.2%)		
Lower left	12 (31.6%)		
Upper left	7 (18.4%)		
Saved gland location (Subtotal parathyroidectomy), number (%)			
Lower right		1 (10.0%)	
Upper right		2 (20.0%)	
Lower left		1 (10.0%)	
Upper left		6 (60.0%)	

**Table 3 medicina-58-01464-t003:** Laboratory findings according to primary and renal (secondary and tertiary) hyperparathyroidism.

Variables	PHPT (n = 37)	RHPT (n = 10)	*p*-Value
PTH, preoperative (pg/mL)	161.6 ± 95.4	1242.1 ± 1075.3	0.011
PTH, pre-incision (pg/mL)	210.3 ± 168.5	1200.4 ± 773.4	0.005
PTH, pre-excision (pg/mL)	143.2 ± 90.8	415.0 ± 269.4	0.056
PTH, 5 min after excision (pg/mL)	64.0 ± 39.2	324.6 ± 258.2	0.087
PTH, 10 min after excision (pg/mL)	39.0 ± 25.0	171.7 ± 138.6	0.021
PTH, 6 months after surgery (pg/mL)	52.1 ± 31.8	71.9 ± 78.2	0.454
PTH, 12 months after surgery (pg/mL)	50.2 ± 27.0	108.7 ± 172.2	0.313
Calcium, preoperative (mg/dL)	10.9 ± 0.9	9.6 ± 1.7	0.042
Calcium, postoperative (mg/dL)	8.9 ± 0.8	7.9 ± 1.9	0.120
Calcium, 6 months after surgery (mg/dL)	9.5 ± 1.3	6.8 ± 1.2	0.003
Calcium, 12 months after surgery (mg/dL)	8.6 ± 2.8	8.6 ± 2.0	0.942
Ionized calcium, preoperative (mmol/L)	1.4 ± 0.1	1.2 ± 0.3	0.132
Ionized calcium, postoperative (mmol/L)	1.1 ± 0.1	1.0 ± 0.2	0.136
Ionized calcium, 6 months after surgery (mmol/L)	1.2 ± 0.1	1.1 ± 0.3	0.334
Ionized calcium, 12 months after surgery (mmol/L)	1.2 ± 0.2	1.1 ± 0.4	0.391
Phosphate, preoperative (mg/dL)	2.6 ± 0.5	3.9 ± 1.1	0.023
Phosphate, postoperative (mg/dL)	3.05 ± 0.9	3.8 ± 1.2	0.061
Phosphate, 6 months after surgery (mg/dL)	3.3 ± 0.7	2.7 ± 0.4	0.075
Phosphate, 12 months after surgery (mg/dL)	3.3 ± 0.6	4.1 ± 1.3	0.377

## Data Availability

The data presented in this study are available in this article.

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
