# Peer review of "Single-Center Experience of Parathyroidectomy Using Intraoperative Parathyroid Hormone Monitoring"

_medicina, 2022, doi:10.3390/medicina58101464_

Round 1

Reviewer 1 Report

The authors reported their experience of parathyroidectomy using IOPTH monitoring. Although the number of participant was relatively small, the importance of IOPTH assay was well-described. I recommend to accept this manuscript.

Author Response

(Ans)

Thank you. We will report a follow-up study when more patient data are accumulated in the future.

Reviewer 2 Report

Dear authors,

The manuscript is well written, but I have some specific comments:

1.     Please add study inclusion/exclusion criteria.

2.     Diagnostic criteria?

3.     Have you encountered any limitations in performing the study?

Author Response

  1. Please add study inclusion/exclusion criteria.

(Ans)

We added the following statement to materials and methods.

(Add)

All patients who underwent parathyroidectomy in Inha University were included and there were no special exclusion criteria.

  1. Diagnostic criteria?

(Ans)

Thanks for pointing it out. We added surgical indications to materials and methods.

(Add)

In PHPT, surgery was performed on patients who had symptoms(nephrolithiasis, fractures, symptomatic hypercalcemia). For asymptomatic PTHP patients, surgery was performed if serum calcium more than 1.0 mg/dL above the normal, creatinine clearance <60 cc/min, nephrocalcinosis or nephrolithiasis identified on imaging, 24-hour urine cal-cium >400 mg/day, osteoporosis by bone density at any site (T score <- 2.5), clinical fragil-ity fracture, vertebral compression fracture on spine imaging, or age <50 years. If medical therapy was refractory, surgery was performed on SHPT patients with symptoms and markedly elevated PTH level. That includes persistent high serum level of intact PTH level >500 pg/m, hyperphosphatemia (serum P>6.0 mg/dL) and/or hypercalcemia (serum Ca >10.0 mg/dL), deformity, fracture, progressive reduction in bone mineral content, ectopic calcification, or neuro-muscular psychiatric symptoms, etc. In THPT, surgery was per-formed on patients with persistent hypercalcemia more than 6 months after kidney trans-plant, low bone mineral density, renal stone or nephrocalcinosis, deterioration of kidney graft due to THPT, symptomatic HPT, or enlarged parathyroid gland detected by US. [2,3]

  1. Have you encountered any limitations in performing the study?

(Ans)

Because the number of HPT patients is relatively small, it will be necessary to analyze more patient data in the future. Further analysis will be conducted when the number of patients gathers. We described this in discussion lines 252-263.